# Beating Carnot efficiency with periodically driven chiral conductors

Sungguen Ryu [1✉], Rosa López [1], Llorenç Serra [1] & David Sánchez [1]

Classically, the power generated by an ideal thermal machine cannot be larger than the Carnot limit. This profound result is rooted in the second law of thermodynamics. A hot question is whether this bound is still valid for microengines operating far from equilibrium. Here, we demonstrate that a quantum chiral conductor driven by AC voltage can indeed work with efficiencies much larger than the Carnot bound. The system also extracts work from common temperature baths, violating Kelvin-Planck statement. Nonetheless, with the proper definition, entropy production is always positive and the second law is preserved. The crucial ingredients to obtain efficiencies beyond the Carnot limit are: i) irreversible entropy production by the photoassisted excitation processes due to the AC field and ii) absence of power injection thanks to chirality. Our results are relevant in view of recent developments that use small conductors to test the fundamental limits of thermodynamic engines.

[1] Instituto de Física Interdisciplinar y Sistemas Complejos IFISC (CSIC-UIB), E-07122 Palma, Spain. ✉email: sungguen@ifisc.uib-csic.es

Quantum thermodynamics is a thriving field that is being nourished by the cross-fertilization of statistical mechanics, quantum information, and quantum transport[1–3]. Interestingly enough, one of its major subjects of enquiry is still connected with the problem that Sadi Carnot analyzed two centuries ago[4], namely, what is the maximum amount of useful work achieved by a generic heat engine? The crucial difference is that the working substance in classical thermodynamic schemes is now a quantum system[5]. This challenges the paradigms of thermodynamics, which are in principle of universal validity, and calls for a revision of our definitions of heat, work, and entropy[6–12].

In general, the efficiency of thermal machines at both macro and nano scales is limited by the entropy production that the second law of thermodynamics dictates to be a positive quantity. In classical thermodynamics the Clausius inequality for entropy production implies a maximum efficiency, the Carnot efficiency. However, this upper bound can in principle be surpassed if we assume quantum coherence is also a resource for entropy production[13,14]. The same argument also implies that a quantum heat engine can violate the Kelvin-Planck statement that no useful work can be extracted from common temperature reservoirs. Hence, understanding how the entropy resource can be controlled in different scenarios is key to achieve enhanced efficiencies in quantum coherent engines and refrigerators.

The widespread interest in quantum thermal machines is also grounded on the plethora of platforms where dynamics can be controlled at a microscopic scale, such as trapped ions[15], quantum dots[16], single-electron boxes[17], optomechanical oscillators[18], QED circuits[19], and multiterminal conductors[20,21]. A shared property of these setups is that the quantum system couples to external baths with which exchanges particles, energy, or different degrees of freedom[22–24]. The baths have thus far been treated with well-defined chemical potentials and temperatures. By contrast, the case of baths that are driven out of equilibrium is quite scarce[25–27]. Such investigation is desirable and timely, as the control and measurement of nanoscale systems driven out of equilibrium by high-frequency AC potential are being experimentally realized, providing single-electron sources. Specifically, we refer to the cases of so-called Levitons[28] in fermionic quantum optics[29,30], quantum-dot pumps for metrology[31], and flying qubits for quantum information processing[32]. The study about heat and energy currents carried by such single-electron sources is of recent interest[33,34].

In this work, we show that a generic class of periodically driven quantum devices can reach thermodynamic efficiencies that surpass the Carnot limit. Our pump engine [see Fig. 1a] consists of a scatterer of arbitrary energy-dependent transmission tunnel coupled to electronic hot and cold reservoirs in the presence of an external AC bias voltage. An AC driving typically generates a finite input power that diminishes the efficiency.

Our key idea to overcome this difficulty is to selectively apply an AC external field to the electrons depending on the direction, which can be implemented using a chiral conductor such as those created with topological matter[35] [see Fig. 1b]. This completely avoids any AC input power, allowing a high efficiency of the quantum engine, in contrast to nonchiral cases. Our main finding is that such a photonic and chiral engine boosts the heat-machine efficiency above the Carnot bound due to a remarkable interplay between nonequilibriumness and chirality. Under these circumstances caution is needed in the thermodynamical description that has to be adapted for a nonthermal bath. We adopt the Floquet scattering matrix approach[36,37] for electric and heat currents and also need a generalized definition of entropy production based on Shannon formula for the incoming and outgoing electron distributions in each terminal[11].

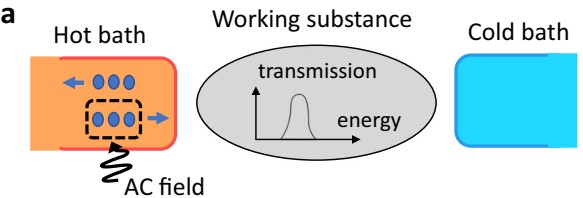

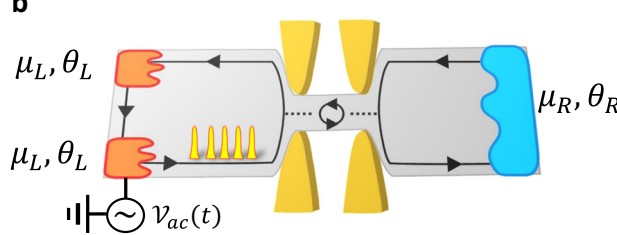

**Fig. 1 Setup. a** Schematic of periodically driven chiral engine. In the hot bath, an AC field is selectively applied only to the electrons directed towards the working substance. The working substance is composed of a scatterer whose transmission probability has an arbitrary energy dependence. **b** An implementation using a chiral conductor with a localized energy level. A time-dependent AC voltage $\mathcal{V}_{ac}(t)$ is applied only to the lower left reservoir, hence realizing the selective AC driving. Edge channels from left and right reservoir are tunnel coupled to the localized state. The hot left reservoirs (cold right reservoir) have temperature $\theta_{L(R)}$ and chemical potential $\mu_{L(R)}$.

## Results

**Floquet scattering matrix.** The setup in Fig. 1b consists of a localized energy level (a quantum dot or impurity) tunnel linked to chiral edge conducting modes with a tunnel rate $\Gamma$. The chiral modes are connected to three reservoirs where the left two reservoirs play the role of the hot reservoir in Fig. 1a with the selective AC driving. This selectivity is realized by applying an AC gate voltage only to the lower reservoir, exploiting the chirality of the conductor. The electrons deep in the left upper or left lower reservoir are described with the Fermi-distribution $f_L(\mathcal{E})$ and the electrons deep in the right reservoir are described with $f_R(\mathcal{E})$, where $f_\beta(\mathcal{E}) = 1/[1 + \exp\{(\mathcal{E} - \mu_\beta)/(k_B \theta_\beta)\}]$ for $\beta = L, R$, $\theta_\beta$ and $\mu_\beta$ are temperature and chemical potential of the reservoir $\beta$, and $k_B$ is Boltzmann constant. We assume $\theta_L \geq \theta_R$. Below, we present the scattering formalism in a two-terminal form, treating the left two reservoirs as a single bath, as we are only interested in the net charge (or heat) currents flowing into the terminals.

In the static case, the scatterer has reflection ($r_{st}$) and transmission ($t_{st}$) amplitudes that depend on energy $\mathcal{E}$:

$$\mathcal{S}^{(st)}(\mathcal{E}) = \begin{bmatrix} r_{st}(\mathcal{E}) & t'_{st}(\mathcal{E}) \\ t_{st}(\mathcal{E}) & r'_{st}(\mathcal{E}) \end{bmatrix}, \tag{1}$$

where the scattering amplitudes are denoted with $'$ when input is from the right reservoir.

In the presence of the external AC bias voltage $\mathcal{V}_{ac}(t)$, whose time average is 0 and frequency is $\Omega/(2\pi)$, an electron of energy $\mathcal{E}$ in the left input reservoir first gains or loses kinetic energy by absorbing ($n > 0$) or emitting ($n < 0$) $|n|$ photons with the transition amplitude $a_n$[38–40],

$$a_n = \frac{\Omega}{2\pi} \int_0^{2\pi/\Omega} dt\, e^{i\phi_{ac}(t)} e^{in\Omega t}. \tag{2}$$

(See Supplementary Information (SI) for a derivation.) $n = 0$

describes the direct process in which photons are neither absorbed nor emitted. $\phi_{ac}(t) \equiv -\int_{-\infty}^{t} dt' e\mathcal{V}_{ac}(t')/\hbar$ is the phase due to the AC voltage. The Floquet scattering matrix[36,37] $\mathcal{S}_{\alpha\beta}(\mathcal{E}_n, \mathcal{E})$ describes the whole process whereby an electron of energy $\mathcal{E}$ from reservoir $\beta$ absorbs/emits $|n|$ photons, scatters off the localized level, and finally enters reservoir $\alpha$ with energy $\mathcal{E}_n \equiv \mathcal{E} + n\hbar\Omega$,

$$\mathcal{S}_{RL}(\mathcal{E}_n, \mathcal{E}) = a_n t_{st}(\mathcal{E}_n), \quad \mathcal{S}_{LL}(\mathcal{E}_n, \mathcal{E}) = a_n r_{st}(\mathcal{E}_n),$$
$$\mathcal{S}_{LR}(\mathcal{E}_n, \mathcal{E}) = \delta_{n,0} t'_{st}(\mathcal{E}), \quad \mathcal{S}_{RR}(\mathcal{E}_n, \mathcal{E}) = \delta_{n,0} r'_{st}(\mathcal{E}). \quad (3)$$

These satisfy unitarity of Floquet scattering matrix, $\sum_{n\alpha}|\mathcal{S}_{\alpha\beta}(\mathcal{E}_n, \mathcal{E})|^2 = 1$, and $\sum_{n\beta}|\mathcal{S}_{\alpha\beta}(\mathcal{E}, \mathcal{E}_{-n})|^2 = 1$ (see SI).

For the results below, it is useful to define the mean number of photons absorbed during the photoassisted scattering process, $\langle n(\mathcal{E}) \rangle_{\alpha\beta} = \sum_n n |\mathcal{S}_{\alpha\beta}(\mathcal{E}_n, \mathcal{E})|^2$. The total transmission probability from the left [right] reservoir, including all photoassisted transitions, becomes $\mathcal{T}(\mathcal{E}) \equiv \sum_n |\mathcal{S}_{RL}(\mathcal{E}_n, \mathcal{E})|^2$ $[\mathcal{T}'(\mathcal{E}) \equiv \sum_n |\mathcal{S}_{LR}(\mathcal{E}_n, \mathcal{E})|^2]$.

**Charge and heat flow and efficiency.** The time-averaged charge, heat, and energy currents flowing into the $\alpha$ reservoir are determined by the above mentioned Floquet scattering matrix as[36]

$$\overline{I_e^\alpha} = e \int_0^\infty \frac{d\mathcal{E}}{h} \sum_{\beta,n} \left\{ -\delta_{\alpha\beta}\delta_{n0} + |\mathcal{S}_{\alpha\beta}(\mathcal{E}_n, \mathcal{E})|^2 \right\} f_\beta(\mathcal{E}), \quad (4)$$

$$\overline{I_h^\alpha} = \int_0^\infty \frac{d\mathcal{E}}{h} \sum_{\beta,n} \{ -\delta_{\alpha\beta}\delta_{n0} + |\mathcal{S}_{\alpha\beta}(\mathcal{E}_n, \mathcal{E})|^2 \}(\mathcal{E}_n - \mu_\alpha) f_\beta(\mathcal{E}), \quad (5)$$

$$\overline{I_u^\alpha} = \int_0^\infty \frac{d\mathcal{E}}{h} \sum_{\beta,n} \{ -\delta_{\alpha\beta}\delta_{n0} + |S_{\alpha\beta}(\mathcal{E}_n, \mathcal{E})|^2 \}\mathcal{E}_n f_\beta(\mathcal{E}), \quad (6)$$

where $e$ ($<0$) is the electron charge and $h$ is the Planck constant. Using unitarity, we obtain

$$\overline{I_e^R} = \frac{e}{h} \int_0^\infty d\mathcal{E} \left\{ \mathcal{T}(\mathcal{E})f_L(\mathcal{E}) - \mathcal{T}'(\mathcal{E})f_R(\mathcal{E}) \right\}. \quad (7)$$

Current conservation, $\overline{I_e^R} + \overline{I_e^L} = 0$, is satisfied over one driving period since no charge piles up in the steady-state. Expanding $\mathcal{E}_n - \mu_\alpha$ into $\mathcal{E} - \mu_\alpha$ and $n\hbar\Omega$, and using again unitarity, we obtain

$$\overline{I_h^R} = \frac{1}{h} \int_0^\infty d\mathcal{E} (\mathcal{E} - \mu_R)\{ \mathcal{T}(\mathcal{E})f_L(\mathcal{E}) - \mathcal{T}'(\mathcal{E})f_R(\mathcal{E}) \}$$
$$+ \frac{\Omega}{2\pi} \sum_{\beta=L,R} \int_0^\infty d\mathcal{E} \langle n(\mathcal{E}) \rangle_{R\beta} f_\beta(\mathcal{E}). \quad (8)$$

$\overline{I_h^L}$ is determined by Eq. (8) with substitutions $(\mathcal{E} - \mu_R) \rightarrow -(\mathcal{E} - \mu_L)$ and $\langle n(\mathcal{E}) \rangle_{R\beta} \rightarrow \langle n(\mathcal{E}) \rangle_{L\beta}$. Note that heat [charge] current is positive when the electrons flow into [out of] the reservoir.

We compute the power associated with DC and AC voltage bias, in order to obtain the net generated power by the chiral conductor $\overline{P_{gen}}$ over a period. The DC voltage bias applied against the current flow generates electrical power $\overline{P_e} = -(\mu_L - \mu_R)\overline{I_e^R}/e$[5]. The time-averaged power injected into the conductor by the AC voltage is related to the energy currents as $\overline{P_{in}} = \overline{I_u^L} + \overline{I_u^R}$ due to energy conservation during one AC period[41,42]. The net generated power is determined by the DC power subtracted by the injected AC power,

$$\overline{P_{gen}} = \overline{P_e} - \overline{P_{in}}, \quad (9)$$

Since the DC power $\overline{P_e}$ is dictated by the electrical flow and the injected AC power by the energy currents, the first law of thermodynamics demands that the net generated power is given

by the heat fluxes as $\overline{P_{gen}} = -(\overline{I_h^R} + \overline{I_h^L})$. Using Eqs. (7)–(9), the input power is written in terms of Floquet scattering matrix as

$$\overline{P_{in}} = \sum_{\alpha\beta} \int \frac{d\mathcal{E}}{h} \hbar\Omega \langle n(\mathcal{E}) \rangle_{\alpha\beta} f_\beta(\mathcal{E}). \quad (10)$$

This clarifies the time-averaged power associated with AC voltage in a general Floquet scattering situation. The term $\hbar\Omega\langle n(\mathcal{E})\rangle_{\alpha\beta}$ describes the energy change of the electron incoming from reservoir $\beta$ and outgoing to $\alpha$ via photon absorptions/emissions. The factor $(d\mathcal{E}/h)f_\beta(\mathcal{E})$ accounts for the injection rate of electrons in the energy window $[\mathcal{E}, \mathcal{E} + d\mathcal{E}]$ from reservoir $\beta$.

A salient feature of our chiral setup is that the power associated with AC voltage is zero, $\overline{P_{in}} = 0$, regardless of the form of the AC driving $\mathcal{V}_{ac}(t)$ and the scatterer. This is a consequence of the fact that the mean photon number $\langle n \rangle \equiv \sum_n n|a_n|^2$ involved in the photon absorption/emission by AC voltage is zero, $\langle n \rangle = 0$ (because $\langle n \rangle = e\overline{\mathcal{V}_{ac}}/(\hbar\Omega) = 0$ as derived in SI) and the Floquet scattering matrix has the form of Eq. (3) for a chiral system. Then, the mean number of absorbed photons by an electron traveling from the left reservoir is also zero, as $\langle n(\mathcal{E})\rangle_{LL} + \langle n(\mathcal{E})\rangle_{RL} = \sum_n n|a_n|^2[|t_{st}(\mathcal{E}_n)|^2 + |r_{st}(\mathcal{E}_n)|^2] = \langle n \rangle$. Besides, an electron from the right reservoir does not absorb nor emit photons, thus $\langle n(\mathcal{E})\rangle_{LR} = \langle n(\mathcal{E})\rangle_{RR} = 0$. Hence, $\overline{P_{in}} = 0$ according to Eq. (10). Contrarily, in the nonchiral case (see SI for details), $\overline{P_{in}}$ is generally positive. In fact, in the limit of slow driving ($\Omega \ll \Gamma/\hbar$) and zero temperature case, the power injection is in the form of Joule's law, $\overline{P_{in}} = |t_{st}(\mu_L)|^2 \frac{e^2}{h} \overline{\mathcal{V}_{ac}^2(t)}$, hence positive. The positive $\overline{P_{in}}$ diminishes the net generated work, and thus efficiency in nonchiral conductors.

When our system satisfies $\overline{I_h^L} < 0$ and $\overline{P_{gen}} > 0$, it works as a heat engine with efficiency $\eta$[41],

$$\eta = \frac{\overline{P_{gen}}}{-\overline{I_h^L}}. \quad (11)$$

In static situations, it is common to compare $\eta$ with the Carnot efficiency $\eta_C = 1 - \theta_R/\theta_L$, i.e., the upper bound dictated by the positivity of the entropy production from Clausius relation,

$$\overline{\dot{S}^{(C)}} = \sum_\alpha \frac{\overline{I_h^\alpha}}{\theta_\alpha}. \quad (12)$$

However, caution is needed for such comparison in nonequilibrium situations. The fact of having nonthermal contacts prevents us from considering the entropy production written as in Eq. (12). Indeed, our results indicate $\overline{\dot{S}^{(C)}} < 0$ for some regimes of parameters (see Fig. 2c), seemingly violating the second law of thermodynamics. To get a deeper insight, we employ the entropy production defined using Shannon entropy of the unperturbed (incoming) distribution function $f_\alpha(\mathcal{E})$ and the AC-driven nonequilibrium distribution function $f_\alpha^{(out)}(\mathcal{E}) = \sum_{n,\beta}|S_{\alpha\beta}(\mathcal{E}, \mathcal{E}_n)|^2 f_\beta(\mathcal{E}_n)$, recently proposed in ref. [11] for a periodically driven system,

$$\overline{\dot{S}} = \frac{k_B}{h} \sum_{\alpha=L,R} \int d\mathcal{E}(-\sigma[f_\alpha(\mathcal{E})] + \sigma[f_\alpha^{(out)}(\mathcal{E})]). \quad (13)$$

Here $\sigma[f] \equiv -f\ln f - (1-f)\ln(1-f)$ is the binary Shannon entropy function measured in nats. A further step clarifies the deviation $\delta\overline{\dot{S}} \equiv \overline{\dot{S}} - \overline{\dot{S}^{(C)}}$ of Eq. (13) from its counterpart deduced from the Clausius relation Eq. (12). Remarkably, we find a simple relation between the deviation and the photon number uncertainty $\delta n \equiv \sqrt{\sum_n (n - \langle n \rangle)^2 |a_n|^2}$, in the regime of small

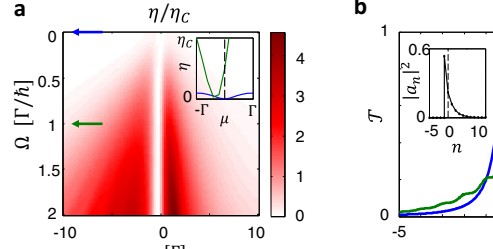

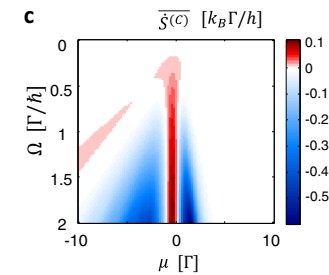

**Fig. 2 Efficiency enhancement beyond the Carnot limit by the AC voltage driving.** Here AC voltage is chosen as periodic Lorentzian pulses which generate Levitons. **a** Efficiency when tuning the driving frequency $\Omega$ and the average chemical potential $\mu \equiv (\mu_L + \mu_R)/2$ (measured from the resonance level). Inset: Plot near $\mu = 0$ (dashed line) for $\Omega = 0$ (blue) and $\Omega = \Gamma/\hbar$ (green) as indicated with arrows in the main panel. **b** Total photoassisted transmission probability $\mathcal{T}$ for an electron of energy $\mathcal{E}$ (measured from the resonance level) incoming from the left reservoir for $\Omega = 0$ (blue) and $\Omega = \Gamma/\hbar$ (green). Inset: the photoassisted transition probabilities $|a_n|^2$. **c** The entropy production rate assuming Clausius relation, $\dot{\overline{S}}^{(C)}$. Here, $(\theta_L + \theta_R)/2 = 0.25\Gamma/k_B$, $\theta_L - \theta_R = 0.1\theta$, $w = 0.05 \times 2\pi/\Omega$, and the DC chemical potential bias is chosen for the maximal power generation.

biases $k_B|\theta_L - \theta_R|$, $|\mu_L - \mu_R| \ll k_B\theta_L$, and small energy uncertainty induced by the AC voltage $\delta n\,\hbar\Omega \ll k_B\theta_L$, (see SI for the details)

$$\delta\overline{S} \approx \frac{(\delta n\hbar\Omega)^2}{2h\theta_L}. \tag{14}$$

The energy uncertainty $\delta n\,\hbar\Omega$ is equal to the root mean square of the AC potential energy, $e(\overline{\mathcal{V}_{ac}^2})^{1/2}$. Hence, Eq. (14) quantifies how the initially thermal electrons in the left reservoir are driven into the nonthermal state due to energy uncertainty induced by the AC voltage.

**Numerical results**. Figure 2 shows the result of our numerical calculations. In the static situation, the scattering matrix $\mathcal{S}^{(st)}$ is modeled as a Breit–Wigner resonance whose full width at half maximum is $\Gamma$. As an illustration showing the most dramatic effects, e.g., dynamical electron-hole symmetry breaking as shown below, of the AC driving, we consider the AC voltage bias of periodic Lorentzian pulses corresponding to the protocol generating Levitons of charge $e$[28,38], $\mathcal{V}_{ac}(t) = \frac{h}{e\pi w}\left[\sum_{m=-\infty}^{\infty}\frac{1}{1+(t-2m\pi/\Omega)^2/w^2}\right] + C$. Here the offset $C$ is chosen to satisfy $\overline{\mathcal{V}_{ac}(t)} = 0$. The temperatures $\theta_L$ and $\theta_R$ are fixed while the average chemical potential $\mu \equiv (\mu_L + \mu_R)/2$ and the AC frequency $\Omega$ are tuned, choosing the DC voltage bias $(\mu_L - \mu_R)/e$ that maximizes the generated power (see SI for details). Figure 2a shows that when the AC driving becomes nonadiabatic, $\hbar\Omega > \Gamma$, the efficiency becomes significantly enhanced with respect to the static cases $\hbar\Omega = 0$. Notably, when the average chemical potential aligns with the resonance level, the driving can even make the system operate as a heat engine while it does not in the static case, hence realizing a photoassisted thermoelectric engine [see inset in Fig. 2a]. This is due to the electron-hole asymmetry dynamically induced by the Levitons. For an AC voltage of a sine-wave form, which does not break electron-hole symmetry, such an effect is not observed. As the frequency increases, the total transmission probability $\mathcal{T}$, see Fig. 2b, shows subpeaks determined by photon-assisted transmissions. For Levitons, the photoassisted transition probability $|a_n|^2$, shown in the inset of Fig. 2b, becomes asymmetric for photon absorption and emission. The analytic expression of $a_n$ is written in the SI, which is experimentally verified by quantum tomography[30].

Remarkably, the efficiencies in the AC-driven cases become larger than the Carnot bound. This behavior is accompanied by the negative entropy production $\overline{S}^{(C)}$ given by the Clausius relation as shown in Fig. 2c. We emphasize that the efficiency enhancement over the Carnot efficiency is not restricted to the

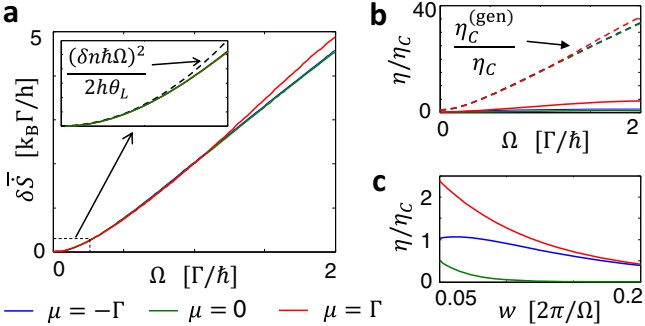

**Fig. 3 Departure from Clausius relation. a** Deviation of the entropy production $\delta\overline{S}$ from Clausius relation, in the situation of Fig. 2. Inset: Comparison with the slow-driving limit, Eq. (14) (dashed line) for $\Omega \in [0, k_B\theta_L/\hbar]$. **b** Efficiency in comparison with the upper bound determined by positivity of entropy production, $\eta_C^{(gen)} = \eta_C + \theta_R\delta\overline{S}/|\overline{I_h^L}|$ (dashed lines). **c** Efficiency when tuning the temporal width $w$ of the Leviton, while fixing $\Omega = \Gamma/\hbar$.

AC voltage of Lorentzian pulses; e.g., we also observe efficiencies beyond Carnot bound for sinusoidal signals.

This seeming violation of the second law of thermodynamics is understood from the fact that the Shannon entropy production Eq. (13) is always nonnegative. Figure 3a shows that the deviation of the entropy production from the Clausius equality, $\delta\overline{S}$, grows as the frequency increases. Upto $\Omega \sim k_B\theta_L/\hbar$, the deviation $\delta\overline{S}$ increases quadratically with respect to $\Omega$, when $w\Omega$ is fixed, independently of the values of $\mu$ as predicted by Eq. (14) because the photon number uncertainty is determined only by the AC voltage driving protocol (see the inset). The positivity of the Shannon entropy production provides a new upper bound $\eta_C^{(gen)} = \eta_C + \theta_R\delta\overline{S}/|\overline{I_h^L}|$ of the efficiency, rather than the Carnot value. Importantly, $\eta_C^{(gen)}$ is not universal unlike $\eta_C$ and depends on the details of the nonequilibrium AC potential. Therefore, there is considerable room to tailor arbitrarily large efficiencies (limited by energy conservation) in AC-driven quantum chiral conductors. The upper bound increases as the frequency increases because $\delta\overline{S}$, hence the nonequilibrium effect, is enhanced [see Fig. 3b]. The anomalous efficiency enhancement by the AC driving becomes stronger when the Lorentzian pulse is more squeezed in the period, as shown by Fig. 3c. This is as expected,

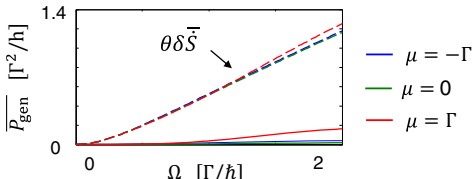

**Fig. 4 Power generation from isothermal baths, $\theta_L = \theta_R = \theta$.** The power generation is allowed upto $\theta \delta \bar{\dot{S}}$ (dashed lines) when the AC driving induces additional entropy production, $\delta \bar{\dot{S}} > 0$. Here $k_B \theta = 0.25\Gamma$.

because when the pulse is more squeezed, the energy uncertainty $\delta n \, \hbar\Omega = e(\overline{\mathcal{V}_{ac}^2})^{1/2}$ becomes larger, hence $\delta \bar{\dot{S}}$ is enhanced.

Figure 4 shows that our engine extracts work even when the temperatures of both reservoirs are equal, $\theta_L = \theta_R = \theta$, violating the traditional Kelvin-Planck statement[43] of the second law of thermodynamics. The power is generated when an electric current is produced by the AC driving and when an electric voltage of a value smaller than a stopping voltage (i.e., a voltage which stops the current) is applied against the current. We find that the electric pumping current has a simple expression when $\delta n \, \hbar\Omega \ll k_B \theta \ll \Gamma$ (see SI for the derivation)

$$\overline{I_e^R}\Big|_{V_L = V_R} = \frac{e(\delta n \hbar \Omega)^2}{2h} \frac{d|t_{st}|^2}{d\mathcal{E}}\Big|_{\mathcal{E} = \mu}. \qquad (15)$$

This is a universal relationship that suggests that the power generation beyond the Carnot limit is possible for any local scatterer of energy-dependent transmission and any AC voltage with nonvanishing fluctuation $\overline{\mathcal{V}_{ac}^2}$. Accordingly, we observe the efficiency enhancement beyond the Carnot limit for the AC voltage of sine-wave signal (not shown) or for a scatterer realized by a quantum point contact[44] (see SI).

One may define an effective temperature[45–47] $\theta_{eff}$ of the left reservoirs, by comparing Eq. (15) with a thermoelectric current in the same situation but substituting the left reservoirs with a static reservoir of temperature $\theta_{eff}$. In this situation, the thermoelectric current satisfies $\overline{I_e^R}\big|_{V_L = V_R} = [e\pi^2 k_B^2 \theta(\theta_{eff} - \theta)/3h](d|t_{st}|^2/d\mathcal{E})\big|_{\mathcal{E} = \mu}$[5], in the same condition used to derive Eq. (15), namely $|\theta_{eff} - \theta| \ll \theta \ll \Gamma/k_B$. This suggests the effective temperature being $\theta_{eff} = \theta + 3(\delta n \hbar \Omega)^2/(2\pi^2 k_B^2 \theta)$, which is manifestly positive. The effective temperature, which measures the broadening of a Fermi-Dirac-like distribution[48], is enhanced due to the AC voltage which rearranges the distribution of electrons in energy in a more uncertain way due to finite $\delta n$, without injecting any net work over a period on the system. However, caution is needed; such an effective temperature does not describe the photoassisted distribution $f_\alpha^{(out)}(\mathcal{E})$ well, because it strongly departs from a Fermi-Dirac distribution when the AC driving becomes nonadiabatic, see Supplementary Fig. 3 in SI. Furthermore, the photoassisted electrons have coherence, i.e., the off-diagonal element in the density matrix in the energy basis, which is known to induce a nontrivial effect in the current noise[49] and cannot be described by the effective temperature. This coherence does not play any role in the time-averaged values of charge and heat current considered here. But it may significantly affect other functionalities of the heat engine, such as time-resolved currents and the thermo-dynamic uncertainty relation[50].

Alternatively, the role of the AC voltage can be interpreted as a nonequilibrium demon[25]. Here, the nonequilibrium demon (AC driving) induces additional entropy production by rearranging the distribution of electrons in energy in a more uncertain way, while satisfying the demon condition: no injection of energy

$\overline{P_{in}} = 0$. Thus, the entropy production deviates from the Clausius equality, and the power generation is allowed upto the upper bound $\theta \dot{S}$. In contrast to the setups of ref. [25], our setup does not need a fine-tuning for the demon condition $\overline{P_{in}} = 0$; the condition is satisfied regardless of the AC voltage profile.

## Discussion

This work demonstrates that an AC-driven chiral conductor can exhibit efficiencies beyond the Carnot limit due to the negative entropy production when the Clausius relation is assumed. To amend the apparent violation of the second law of thermo-dynamics we employ a positively defined entropy production based on the Shannon entropy applied for the incoming and outgoing electronic distribution functions of the reservoirs. Interestingly, we find that the deviation of entropy production from the Clausius relation is given by the photon number uncertainty of the AC driving. Chiral transport is crucial for efficiency enhancement. Nonchiral conductors do not exhibit efficiencies beyond the Carnot limit due to the finite injection energy which diminishes the generated power. The regime for achieving efficiency beyond the Carnot limit, $\Omega > \Gamma/h$, is realistic because with the state-of-the-art fast AC voltage of Lorentzian pulses of width $w = 15$ ps with AC period $2\pi/\Omega = 166$ ps (or sinusoidal signal with AC period of 42 ps)[29], the regime $\Omega > \Gamma/h$ is approachable as long as the level broadening is small enough as $\Gamma < 0.025$ meV (or $\Gamma < 0.1$ meV for the sinusoidal signal), which can be tuned with the tunnel coupling via finger gates.

## Data availability

The authors declare that all data supporting the findings of this study are available within the article.

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

## Acknowledgements

We acknowledge support from Grants No. MAT2017-82639 and No. PID2020-117347GB-I00 funded by MCIN/AEI/10.13039/501100011033, No. MDM2017-0711 funded by MINECO/AEI/FEDER María de Maeztu Program for Units of Excellence, and No. PDR2020-12 funded by GOIB. S.R. also acknowledges partial support from National Research Foundation of Korea (Grant No. 2021R1A6A3A03040076).

## Author contributions

All authors, S.R., R.L., L.S., and D.S., conceived the project. S.R. performed the calculations. All authors discussed the results and made significant contributions to the manuscript.

## Competing interests

The authors declare no competing interests.
