## [Peer Review File · Nature Communications]

Title: Beating Carnot efficiency with periodically driven chiral conductorsREVIEWER COMMENTS

Reviewer #1 (Remarks to the Author):

The Authors study a periodically driven chiral quantum conductor aimed at realizing thermal quantum engine. Taking advantage of the chirality which makes the ac power injection vanishing and of the particular entropy production given by the photo-assisted processes, they show that the Carnot efficiency can be enhanced and that work can be extracted without the need of two different reservoir temperature. For their demonstration, the Authors considered a very specific example, a conductor made of a single localized state connecting the two reservoirs. While we do not questioned the validity of the result, we think the work fails to provide the reader with a convincing way to identify the general quantum principles from which the classical Carnot efficiency relation is violated. The work provides novel material. However, as such, the work is not publishable and needs considerable revision.

Here are some important remarks:

Figure 1 is puzzling. A chiral conductor is made of four distinct reservoirs, but only two injecting reservoirs (lower left and upper right) are considered in the work. The figure suggests that the left and right reservoirs absorbing the current have the chemical potential and temperature of the left and right emitting reservoirs respectively. In practice many situation may occur. If, for example the upper left contact is grounded, some energy will be dissipated in it, if it is floating, photo-assisted processes will generated extra energy into the left emitting contact. All these situations should be discussed and clarified to give a complete picture.

An important figure is also missing. The reader would appreciate a figure where one can clearly identify the non-equilibrium resource part, the working part and the interface in between. This is implicit in the text but buried into too many definitions confusedly telling into where or from which part the energy flows.

Regarding the definition of a useful work that extractable from the thermal engine, the reader would be satisfied to see a connection with the thermo-electric current generated by the combination of energy dependent transmission and photon-assisted processes.

Having an energy dependent transmission is a key feature for electrical heat engine. This is nowhere discussed. In the present work, this implicitly enters into the problem via the numerical treatment of the specific single energy level which makes the interface between the four reservoirs. A discussion of the need for a scatter with energy-dependent transmission (or including an extra example) would be appreciated. For example, a QPC whose transmission obeys a Fermi-Dirac-like dependence in energy.

Here are some extra suggestion to improve the readability:

Why not introduce the energy current $I_{\{u\}}$ together with the heat and electrical current, equation (4) and (5) by simply using $I_{\{h\}} = I_{\{u\}} - \mu_{\{\alpha\}} I_{\{e\}}^{\{\alpha\}}$ and put their expression in terms of

Floquet matrix into a full in-line equation

In Fig.2(a) inset. The zero value of η occurs for negative μ , green curve. This is specific to Lorentzian pulses, which give asymmetric Floquet coefficients in energy. If sine-wave pulses were used this would occur for $\mu=0$. This may be mentioned, perhaps in combination with the specific energy dependence of the transmission. Would this occur for the stepwise energy dependence of a QPC?

In the text, right column page 3, the definition of the non-equilibrium distribution f_{out} requires some warning. It is well known that this photo-assisted distribution cannot be considered as pure distribution because it is the diagonal part of an energy density matrix which contains non-zero diagonal terms. It may be dangerous to use it in place of a thermal distribution. Indeed, it is well known, and experimentally proven that this distribution remarkably leads to zero current noise for a perfectly transmitted channel while a thermal distribution leads to finite Johnson-Nyquist noise. The role of quantum coherence, which is a key ingredient for enhancement of Carnot efficiency, should be discussed.

Reviewer #3 (Remarks to the Author):

The authors study efficiency of a heat engine based on periodically driven Chiral conductors and show that it can beat the universal Carnot bound. Further, it is shown that work can be extracted from common temperature baths. They also show that the entropy production is still positive and second law of thermodynamics is not violated. The paper is written well and the message is interesting because of the recent quest to go beyond the performance of conventional heat engines using the principles of quantum theory. I would recommend the article for publication if the following concerns are addressed satisfactorily.

The incoming and outgoing distributions of left and right reservoirs can be non-equilibrium distributions when the system is working as a heat engine. Is it possible to find an effective temperature corresponding to these four non-equilibrium distributions? Now, out of four effective temperatures, taking the highest and lowest temperatures, one can define a new Carnot efficiency. Does the Carnot efficiency defined by these effective temperatures lies above the efficiency of the system?

There is a typo in the page 2 (line 94) where Ω is defined.

Reply to Reviewer #1

We thank the Reviewer for the positive assessment of our work and for the detailed report. Below, we quote the Reviewer's comments in italics.

Figure 1 is puzzling. A chiral conductor is made of four distinct reservoirs, but only two injecting reservoirs (lower left and upper right) are considered in the work. The figure suggests that the left and right reservoirs absorbing the current have the chemical potential and temperature of the left and right emitting reservoirs respectively. In practice many situation may occur. If, for example the upper left contact is grounded, some energy will be dissipated in it, if it is floating, photo-assisted processes will generated extra energy into the left emitting contact. All these situations should be discussed and clarified to give a complete picture.

An important figure is also missing. The reader would appreciate a figure where one can clearly identify the non-equilibrium resource part, the working part and the interface in between. This is implicit in the text but buried into too many definitions confusedly telling into where or from which part the energy flows.

We find this comment quite illuminating. For a better description of our proposal, we have added in the revised manuscript the new Fig. 1(a), showing the generic components of our system. It consists of an energy-dependent transmission scatterer connected to one hot and one cold baths. This is the prototypical structure for a heat engine. Our key ingredient is an AC field that selectively applies only to the hot-bath electrons propagating towards the scatterer. This selective AC field can be implemented by using a chiral conductor as shown in new Fig. 1(b). As a consequence, only a terminal is needed for the cold bath (and not two as in the previous version). This is the minimal setup that realizes the scheme of Fig. 1(a).

We agree with the Reviewer that in a multiterminal device many situations can occur. However, our choice for the reservoir configuration where the left two reservoirs have the same temperature and chemical potential is justified because they together form the hot bath in the scheme of Fig. 1(a).

We have clarified this issue in the revised manuscript (see both the new text in the first paragraph of p. 2 and the caption of Fig. 1).

Finally, we attach a discussion at the end of this reply, where we consider the case when the chemical potentials of the left two reservoirs are different. This result shows that a potential bias between the two induces a power dissipation, as correctly pointed out by the Reviewer. Since this effect is detrimental to the power generation, our choice of equal chemical potentials maximizes the efficiency, which is the goal of the present manuscript. Note that we do not include this discussion in the manuscript for better readability, but we could do it if the Reviewer thinks differently.

Regarding the definition of a useful work that extractable from the thermal engine, the reader would be satisfied to see a connection with the thermo-electric current generated by the combination of energy dependent transmission and photon-assisted processes.

Inspired by this comment and a remark from Reviewer #3, we have now clarified the connection among the power generated by our engine, the energy-dependent transmission, and the photon-assisted processes. We have added a new paragraph which includes new Eq. (15).

Having an energy dependent transmission is a key feature for electrical heat engine. This is nowhere discussed. In the present work, this implicitly enters into the problem via the numerical treatment of the specific single energy level which makes the interface between the four reservoirs. A discussion of the need for a scatter with energy-dependent transmission (or including an extra example) would be appreciated. For example, a QPC whose transmission obeys a Fermi-Dirac-like dependence in energy.

We agree with the Reviewer that having an energy dependent scatterer is a salient feature of our heat engine. It turns out that there exists a universal relation, given by new Eq. (15), which shows that power generation beyond the Carnot limit is possible for any local scatterer with energy-dependent transmission and any AC voltage with nonvanishing fluctuation [see the new paragraph including new Eq. (15)]. We have also emphasized the importance of energy-dependent transmissions by explicitly depicting it in the new setup of Fig. 1(a).

Additionally, we have performed numerical calculations when the engine scatterer is a QPC instead of a quantum dot. The results are discussed in the new section "ENGINE WITH A QUANTUM POINT CONTACT" inside the SM. In agreement with new Eq. (15), the efficiency can also be enhanced beyond the Carnot limit (see Fig. S4). We have also compared, in the same SM section, the QPC and the quantum dot cases.

Here are some extra suggestion to improve the readability:

Why not introduce the energy current $I_{\{u\}}$ together with the heat and electrical current, equation (4) and (5) by simply using $I_{\{h\}} = I_{\{u\}} - \#mu_{\{\alpha\}} I_{\{e\}}^{\{\alpha\}}$ and put their expression in terms of Floquet matrix into a full in-line equation.

We have revised the manuscript as suggested, adding new Eqs. 4-6.

In Fig.2(a) inset. The zero value of $\#eta$ occurs for negative $\#mu$, green curve. This is specific to Lorentzian pulses, which give asymmetric Floquet coefficients in energy. If sine-wave pulses were used this would occur for $\#mu=0$. This may be mentioned, perhaps in combination with the specific energy dependence of the transmission. Would this occur for the stepwise energy dependence of a QPC?

Indeed, the efficiency η becomes zero for $\mu=0$ if a sine-wave signal is used instead of the Lorentzian pulses for the AC voltage. In other words, the AC driving using sine-wave signal does not generate a heat engine at $\mu=0$, because of it does not induce any electron-hole asymmetry.

We have added this remark in the revised manuscript (see the first paragraph in the section Results in p. 4).

For the case of the engine with QPC at $\mu=0$, there is no direct analogy to the quantum dot case because the efficiency is already finite in the absence of the AC driving due to the asymmetry of the transmission function.

In the text, right column page 3, the definition of the non-equilibrium distribution f_{out} requires some warning. It is well known that this photo-assisted distribution cannot be considered as pure distribution because it is the diagonal part of an energy density matrix which contains non-zero diagonal terms. It may be dangerous to use it in place of a thermal distribution. Indeed, it is well known, and experimentally proven that this distribution remarkably leads to zero current noise for a perfectly transmitted channel while a thermal distribution leads to finite Johnson-Nyquist noise. The role of quantum coherence, which is a key ingredient for enhancement of Carnot efficiency, should be discussed.

We agree that quantum coherence may play a relevant role in the working operation of our heat engine. However, coherence due to superposition of the photoassisted processes does not influence the time-averaged values of charge current, heat current, power, and efficiency, which are the observables computed in the present work. We think that the coherence may interestingly affect time-dependent currents as in, e.g., the thermodynamic uncertainty relation that connects the electric current noise (or equivalently the power fluctuation) and the entropy production (or equivalently the efficiency). This is an interesting research question to be pursued in the future.

To clarify this point, we have added a discussion in the revised manuscript (see the first paragraph in p. 5).

Regarding the concern about the quantity f_{out} , we would like to point out that we only introduced f_{out} for the Shannon entropy flow, new Eq. (13), in order to understand the efficiency enhancement above the Carnot limit. Our results for the charge current, heat current, power, and efficiency beyond the Carnot limit are unaffected by f_{out} .

DISCUSSION ON RESERVOIRS WITH DIFFERENT CHEMICAL POTENTIALS

In response to Reviewer #1, we discuss in detail a reservoir configuration, different from the one used in Fig. 1(b), that involves power dissipation. We consider a setup where each reservoir can have arbitrary chemical potentials, see Fig. R1. As in the main text, we employ the Floquet scattering formalism to calculate the currents into all the terminals. Then we show that any difference in the chemical potentials between the left two reservoirs diminishes the power generation as the potential bias between the two induces a power dissipation. This justifies our choice for the reservoir configuration in Fig. 1(b).

FIG. R1. Three terminal setup with arbitrary chemical potentials for each reservoir.

The Floquet scattering matrix represented in the reservoir indexes ordered as L_1 , L_2 , and R is

$$S_{\alpha\beta}(\mathcal{E}_n, \mathcal{E}) = \begin{bmatrix} 0 & a_{-n}^* & 0 \\ a_n r_{st}(\mathcal{E}_n) & 0 & \delta_{n0} t'_{st}(\mathcal{E}) \\ a_n t_{st}(\mathcal{E}_n) & 0 & \delta_{n0} r'_{st}(\mathcal{E}) \end{bmatrix}. \quad (\text{R1})$$

The element $S_{L_1 L_2}(\mathcal{E}_n, \mathcal{E})$ is a_{-n}^* because the electron enters into the region under the AC voltage (see Section PHOTOTRANSITION AMPLITUDE a_n in SM). This term does not contribute to the net heat and charge current, $\overline{I}_h^L = \overline{I}_h^{L_1} + \overline{I}_h^{L_2}$, \overline{I}_h^R , and \overline{I}_e^R . We note that Eq. (R1) satisfies the unitarity conditions, $\sum_{n\alpha} |S_{\alpha\beta}(\mathcal{E}_n, \mathcal{E})|^2 = 1$ and $\sum_{n\beta} |S_{\alpha\beta}(\mathcal{E}, \mathcal{E}_{-n})|^2 = 1$.

The charge, heat, and energy currents into reservoir α are determined by Eq. (R1) and Eqs. (4)-(6). The power injection by AC voltage is obtained using the energy conservation relation, $\overline{P}_{in} = \sum_{\alpha} \overline{I}_u^{\alpha}$,

$$\overline{P}_{in} = \int \frac{d\mathcal{E}}{h} \sum_{\alpha, \beta, n} n \hbar \Omega |S_{\alpha\beta}(\mathcal{E}_n, \mathcal{E})|^2 f_{\beta}(\mathcal{E}). \quad (\text{R2})$$

This vanishes due to the chirality manifested in Eq. (R1), $\sum_n |a_n|^2 = 1$, and $\sum_n n |a_n|^2 = 0$,

$$\overline{P}_{in} = 0. \quad (\text{R3})$$

The electric power generated by the charge flow is

$$\overline{P}_e = \sum_{\alpha} \mu_{\alpha} \overline{I}_e^{\alpha} / e. \quad (\text{R4})$$

Using the form of the Floquet scattering matrix, Eq. (R1), we obtain the electric power,

$$\begin{aligned} \overline{P}_e &= (\mu_R - \mu_{L_2}) \frac{\overline{I}_e^R}{e} \\ &+ (\mu_{L_1} - \mu_{L_2}) \int \frac{d\mathcal{E}}{h} \left(-f_{L_1}(\mathcal{E}) + f_{L_2}(\mathcal{E}) \right), \end{aligned} \quad (\text{R5})$$

The second term is a nonpositive quantity which describes the power dissipation between the reservoirs L_1 and L_2 . It vanishes when the two reservoirs have the same chemical potential.

Therefore, the condition used in the paper, $\mu_{L_1} = \mu_{L_2}$, is the best choice for obtaining large electric power. Further, defining $\mu_L \equiv \mu_{L_1} = \mu_{L_2} \equiv$ and $f_L(\mathcal{E}) \equiv f_{L_1}(\mathcal{E}) = f_{L_2}(\mathcal{E})$, the electric power \overline{P}_e , the total heat current into the hot reservoir $\overline{I}_h^L \equiv \overline{I}_h^{L_1} + \overline{I}_h^{L_2}$, and the efficiency become equal to the quantities described in the manuscript.

Reply to Reviewer #3

We thank the Reviewer for the recommendation of publication. Below, we address the concerns.

The incoming and outgoing distributions of left and right reservoirs can be non-equilibrium distributions when the system is working as a heat engine. Is it possible to find an effective temperature corresponding to these four non-equilibrium distributions? Now, out of four effective temperatures, taking the highest and lowest temperatures, one can define a new Carnot efficiency. Does the Carnot efficiency defined by these effective temperatures lies above the efficiency of the system?

This comment is quite interesting. We have carefully investigated the possibility of defining an effective temperature, based on related works such as new Refs. [43-45]

Inspired by this comment and similar views from Reviewer #1, we have found a new Eq. (15) which determines the electric pumping current in terms of the energy-sensitivity of the transmission and the energy uncertainty induced by the AC driving. See also the discussion that follows new Eq. (15). This equation suggests that the left reservoirs may be described by an effective temperature which increases from the original temperature due to the energy uncertainty induced by the AC driving.

However, it is not possible to define a new Carnot efficiency determined by this effective temperature. The underlying reason is that the effective temperature cannot describe well the photoassisted distribution $f_{\omega\alpha}^{\text{(out)}}$, since $f_{\omega\alpha}^{\text{(out)}}$ departs from a Fermi-Dirac distribution. As shown in new Fig. S4 in the revised SM, the photoassisted distributions have many details that cannot be accounted for by thermal broadening. The effective temperature is indeed a useful concept, as illustrated by new Eq. (15), but it cannot describe quantum coherence either.

To clarify the above points, we included a new discussion in the revised manuscript in the first paragraph after new Eq. (15) and added a new figure Fig.S3 in Supplemental Material.

There is a typo in the page 2 (line 94) where ω is defined.

Thanks for pointing it out! The typo is now corrected.

REVIEWER COMMENTS

Reviewer #1 (Remarks to the Author):

The authors have convincingly answered to the Referees questions and have made corresponding amendments in the manuscript.

To my opinion, the new manuscript as reach the level for publication in a high level review as Nature Communication.

I recommend the manuscript be published as is although provided the authors put the discussion at the end of the answer to Referee#1 in a supplementary material.

Reviewer #3 (Remarks to the Author):

The authors have addressed the previous questions and made necessary changes in the text.

In the new version following Eq. (15), an effective temperature in the presence of the drive is described which is greater than the equilibrium temperature of the reservoir. In other words, average energy of the distribution can be different in the presence of the drive (if one defines effective temperature as a measure of average energy [Phys. Rev. Lett. 92, 056804 (2004)]). Also from Fig. (S3), it seems that the average energy of the system (from non-equilibrium distributions) is different due to the driving. I would suggest that authors need to clarify this point even since $P_{in}=0$ (as shown in the manuscript). Usually in open quantum systems, work done from the drive creates coherences in the density matrix. If there is a way of injecting average energy into the system from ac drive (work done on the system), one cannot use a stronger statement (without accounting for the work done on the system) as beating Carnot efficiency, (or violating Kelvin-Planck statement) rather it needs to be addressed as apparent effects. I would suggest authors to dilute the claims about surpassing the Carnot bounds (or Violating Kelvin-Planck statement) accordingly if there is a work done on the system due to the drive.

Though the paper has many interesting results and I would like to recommend this for publication if the above issue is addressed.

Reply to Reviewer #1

We thank the Reviewer for recommending our work for the publication in Nature Communications. We revised our Supplemental Material, following his/her suggestion to include the discussion on reservoirs with different chemical potential presented in the previous reply into the Supplemental Material.

Reply to Reviewer #3

We thank the Reviewer for positive evaluation of our work. Below, we address the concern.

In the new version following Eq. (15), an effective temperature in the presence of the drive is described which is greater than the equilibrium temperature of the reservoir. In other words, average energy of the distribution can be different in the presence of the drive (if one defines effective temperature as a measure of average energy [Phys. Rev. Lett. 92, 056804 (2004)]). Also from Fig. (S3), it seems that the average energy of the system (from non-equilibrium distributions) is different due to the driving. I would suggest that authors need to clarify this point even since $\mu_{in}=0$ (as shown in the manuscript).

We appreciate this suggestion for further clarification. We would like to clarify that the effective temperature, suggested in our work and defined similarly in the work of *Phys. Rev. Lett. 92, 056804 (2004)*, measures the broadening of a Fermi-Dirac-like distribution, rather than the average energy of the system. Please note that the Eq. 9 in *Phys. Rev. Lett. 92, 056804 (2004)*

$$k_B T_{\text{eff}} = \frac{\sqrt{6}}{\pi} \sqrt{\int_{-\infty}^{\infty} [f(E) - 1 + \theta(E)] E dE},$$

which defines an effective temperature T_{eff} for a nonequilibrium distribution $f(E)$ having chemical potential at 0 (where $\theta(E)$ is Heaviside step function), does not measure an average energy because the term $f(E) - 1 + \theta(E)$ can take negative values and cannot thus represent a probability distribution. Therefore, an effective temperature understood as a broadening can increase without any net work over a period done on the system by the AC driving. This is indeed what happens in our chiral heat engine. The effective temperature of the electrons in the left input reservoir is enhanced after photon absorption/emission due to the AC voltage, as the AC voltage rearranges the distribution of electrons in energy in a more uncertain way due to the uncertainty δn of the photon number and the effective temperature is given by $\theta_{\text{eff}} = \theta + 3(\delta n \hbar \Omega)^2 / (2\pi^2 k_B^2 \theta)$ as described in our manuscript. At the same time, the AC voltage does not inject any net work over a period on the system because the average photon number is 0. This is illustrated in, e.g., the inset of Fig. 2(b).

We revised our manuscript, adding both a comment that clarifies this point in the paragraph below Eq. (15) and a citation to the Reviewer's suggested paper.

Usually in open quantum systems, work done from the drive creates coherences in the density matrix. If there is a way of injecting average energy into the system from ac drive (work done on the system), one cannot use a stronger statement (without accounting for the work done on the system) as beating Carnot efficiency, (or violating Kelvin-Planck statement) rather it needs to be addressed as apparent effects. I would suggest authors to dilute the claims about surpassing the Carnot bounds (or Violating Kelvin-Planck statement) accordingly if there is a work done on the system due to the drive.

We agree that if additional sources of work are present in the system the Kelvin-Planck statement is trivially violated. However, we would like to emphasize that this violation occurs in our system despite the fact that the net work done by the AC drive is zero. This is an unexpected result due to a nice combination of chirality and quantum scattering.

Please notice also that there is a sentence in the conclusions of the paper emphasizing that our claim on violation of Carnot bounds does not apply in cases of finite injection energy: "Nonchiral conductors do not exhibit efficiencies beyond the Carnot limit due to the finite injection energy which diminishes the generated power.

REVIEWERS' COMMENTS

Reviewer #3 (Remarks to the Author):

Authors addressed previous comments satisfactorily. The results of the paper are interesting and may invoke some discussions on the performance of quantum thermal devices. I do not have further comments and I recommend this article for publication.